# The Effectiveness of Sensory Integration Interventions on Motor and Sensory Functions in Infants with Cortical Vision Impairment and Cerebral Palsy: A Single Blind Randomized Controlled Trial

**DOI:** 10.3390/children9081123

**Published:** 2022-07-27

**Authors:** Mustafa Cemali, Serkan Pekçetin, Esra Akı

**Affiliations:** 1Department of Occupational Therapy, Faculty of Health Sciences, Hacettepe University, Ankara 06050, Turkey; esraaki@hotmail.com; 2Department of Occupational Therapy, Faculty of Gülhane Health Sciences, University of Health Sciences Turkey, Ankara 06018, Turkey; serkanpekcetin@gmail.com

**Keywords:** cerebral palsy, cortical visual impairment, sensory integration, motor function, early intervention

## Abstract

Cortical vision impairment (CVI) and Cerebral Palsy (CP) lead to decrement in sensory and motor functions of infants. The current study examined the effectiveness of sensory integration interventions on sensory, motor, and oculomotor skills in infants with cortical vision impairment. Thirty-four infants with and CP aged 12–18 months were enrolled to the study. The infants were randomly divided into two groups as the control and intervention groups. The intervention group took sensory integration intervention 2 days a week for 8 weeks in addition to conventional physiotherapy 2 days a week for 8 weeks. The control group only received the conventional physiotherapy program 2 days a week for 8 weeks. The duration of the treatment sessions were 45 min for both interventions. Before and after the intervention, sensory processing functions were evaluated with the Test of Sensory Functions in Infants (TSFI), and motor functions were evaluated with the Alberta Infant Motor Scale (AIMS). There was a statistically significant difference between the pre- and post-test mean TSFI total and AIMS scores in the intervention group and control group (*p* < 0.001). The intervention group mean TSFI scores were more statistically significant than the those of the control group. Mean post-intervention AIMS scores did not differ between groups. Sensory integration intervention delivered with the conventional physiotherapy program was more effective than the conventional physiotherapy program in increasing sensory processing skills in one measure in infants with CVI and CP.

## 1. Introduction

Cortical vision impairment (CVI) is generally referred as a visual loss that occurs in the lateral geniculate nucleus (LGN) and the structures following LGN, which cannot be explained by the damage in the eyeball and the optic nerve [1]. The main disorders that cause CVI are hypoxic ischemic encephalopathy, epilepsy, focal brain lesions (including vascular anomalies and intracranial haemorrhages), central nervous system infections (such as meningitis and encephalitis), hydrocephalus, head trauma, new-born hypoglycaemia, pathological or genetic brain anomalies, metabolic diseases, autism spectrum disorder, and cerebral palsy [2].

In infants, CVI often accompanies cerebral palsy (CP) [3]. Cortical vision loss is seen in 60–70% of children with CP. One of the most common problems in children with CP is CVI [4]. In CVI, limitations related to visual acuity, visual field, contrast sensitivity, colour vision, motion perception, and oculomotor mobility can be seen clinically [5]. Sensory and motor functions of the eye, processing of information from the body to the brain, eye-body movement coordination, and sensory integration are impaired due to CVI. Due to this problem, infants experience problems in environmental orientation and body perception, and motor limitations accompany this situation, so infants are less exposed to sensory stimuli. This loss of vision at the cortical level causes sensory–perception–motor problems in addition to the motor function loss of CP [6].

Children with CP experience sensory processing difficulties in visual auditory vestibular tactile position when compared with healthy children [6]. In a randomized controlled study conducted with 22 children aged 2–6 years with spastic CP who did not experience CVI, it was found that sensory integration therapy applied 3 days a week for 3 months improves motor skills [7].

Previous researchers have shown the efficacy of sensory integration therapy on fine motor skills [8] and walking parameters [9] in children with CP who did not experience CVI. In a randomized controlled study with 26 children aged 2–4 years with CVI and spastic CP, it was found that sensory integration therapy applied in addition to conventional physiotherapy for 3 days a week for 3 months improved gross and fine motor skills [10].

The literature showed that visual, physical, cognitive, and behavioural trainings are generally administered in individuals with CVI [11,12,13]. Sensory processing problems are experienced in children with CVI [14]; however, to our knowledge, there is no sensory integration intervention study in children with CVI. When studies with infants and children with CP were examined, it was observed that the literature has mainly included physical, cognitive, and language-speech assessment studies and intervention studies [15,16,17]. There are also sensory integration assessment and intervention studies [6,18,19,20]. There are limited studies in the literature on sensory integration assessment and intervention in infants with CVI and CP. Therefore, our study, which we think will fill an important gap in this area, aimed to assess the effect of sensory integration intervention on sensory processing and motor skills in infants with CVI and CP. The hypothesis of the study is that sensory integration therapy improves sensory and motor skills in infants with cerebral palsy with cortical visual impairment.

## 2. Materials and Methods

The current study was designed according to the CONSORT statement, which provides a standardized method for RCT designs [21]. The local ethical committee approved the study protocol (Registration number: NCT05431647, Approval Date: 17 December 2020), and the study was carried out at the Private Special Education and Rehabilitation Centre between January and June in 2021. Written informed consent was obtained from the families.

### 2.1. Participants

Power analysis was performed to calculate the sample size required to detect a significant effect size (Cohen’s d = 0.80) of the groups on the TSFI. Seventeen subjects were needed in each group to ensure a power of 80%, assuming a two-tailed test for α = 0.05. Thirty-six infants with CP and CVI, aged 12–18 months, were included in the study. The mean age of the infants was 14.47 ± 1.28 months in the intervention group and 13.82 ± 1.55 months in the control group. Two infants were excluded from the study before group allocations because their parents did not want to continue the study. The patients were divided into 2 groups in a randomized method. The study was completed with 34 children with CP and CVI (Figure 1).

Infants with CP and CVI and without hearing loss, congenital anomalies, or systemic disease were included in the study. No distinction was made between the types of CP. Those who received physiotherapy training previously were not excluded from the study, but those who received sensory integration training was excluded from the study. The participants were randomized (allocation ratio of 1:1) to either the intervention or the control group using computer-generated randomization. Both groups were given physiotherapy training as 2 sessions of 45 min per week for 8 weeks. In addition to the physiotherapy training, the intervention group received sensory integration training as 2 sessions of 45 min per week for 8 weeks. The Test of Sensory Functions in Infants and the Alberta Infant Motor Scale Sensory were administered before and after the 8-week intervention. All interventions and assessments were performed by a physiotherapist who worked in the field of physiotherapy for 8 years and occupational therapy for 6 years, blind to assessment results. This physiotherapist has a master’s degree in both fields and is currently pursuing a doctorate in occupational therapy.

### 2.2. Instruments

#### 2.2.1. Test of Sensory Functions in Infants (TSFI)

The TSFI consists of 24 items. It was developed to evaluate sensory processing problems in infants aged 4 to 18 months. The test consists of 5 subsections and 24 items. The subsections of the test are tactile deep pressure response, adaptive motor functions, visual–tactile integration, oculomotor, and response to vestibular stimuli. The subsections evaluate tactile processing, motor praxis, integrated response of the visual and tactile systems, ocular movements, and vestibular processing, respectively. The test requires the infant to be stimulated and interacted with various materials, and the infant’s responses are observed and scored by the clinician. The total score ranges from 0 to 49, with higher scores indicating better sensory processing. The test has cut-off values for both the total score and the subsections according to different age groups. Using these values, sensory processing ability is evaluated as normal, risky, or abnormal [22]. The Turkish validity and reliability study of the TSFI was conducted in 2014. The Cronbach Alpha coefficient of the scale was calculated as 0.875 [23].

#### 2.2.2. Alberta Infant Motor Scale (AIMS)

The AIMS gives information about the gross motor development of infants whose corrected ages are between 0 and 18 months. In AIMS, the age of infants is calculated as the corrected age. The corrected age is calculated according to the 40-week period that the infant must spend in the womb. The corrected age is obtained by subtracting the number of weeks that the infant was born prematurely from 40 weeks (gestational age of term infants), and then this number is subtracted from the infant’s chronological postnatal age [24]. It allows the family and the clinician to obtain information about the infant’s current motor development and to compare the motor development before and after treatment. The test is based on observing the infant’s spontaneous movements. It consists of 4 sub-sections, these being prone, supine, sitting, and standing, and 58 items. For each item, it is determined whether the infant performs that item by paying attention to postural smoothness, antigravity movements, and contact surface with the ground. It is summed up by giving a score of one for each item they successfully perform, and zero for each item they fail to perform. At the end of the evaluation, the total score is obtained by summing the scores of the 4 subsections, and this score is converted to the percentile score showing the infant’s status relative to their peers. The reliability of the AIMS, which has an internal correlation coefficient between 0.96 and 0.98, is high [25].

### 2.3. Intervention

The intervention duration and protocol were designed according to the one of our previous works [26]. The intervention program includes conventional physiotherapy and sensory integration training. Both control and intervention groups were applied physiotherapy intervention as 2 sessions of 45 min per week for 8 weeks. In addition to the physiotherapy intervention, the intervention group received sensory integration intervention as 2 sessions of 45 min per week for 8 weeks. Sensory integration therapy was performed on 2 separate days in the same week that there was no physical therapy session. Conventional physiotherapy applied to both groups included classical physiotherapy, such as rotation, sitting without support, standing and balance and strengthening exercises. Sensory integration therapy applied only to the intervention group included interventions involving vision, hearing, touch, and vestibular stimuli. Sensory integration the intervention was planned considering the TSFI test results. In the TSFI assessment, each subtest has a cut-off score. If the infant’s subtest score is below this value at the end of the evaluation, it indicates that there is a problem in that sensory area. In the current study, the researchers established the sensory intervention plan by considering the cut-off scores in the evaluation of these areas. We planned the details of this intervention according to the infant’s response to stimuli in each item in the TSFI assessment. This intervention included tactile, vestibular, proprioceptive, visual, and auditory stimuli. In this intervention, different patterned fabrics, plushies, and toys with different surfaces for the tactile stimulus; swings and exercise balls for vestibular stimulus; positioning and approximations for proprioception; for visual studies, activities, such as eye tracking in different directions, were carried out with toys of different colours and sizes.

The treatment room was designed according to Parham’s principles of sensory integration therapy. An individualized sensory integration intervention based on the basic principles of sensory integration therapy developed by Parham was applied [27]. According to the evaluations, infants may have problems in different sensory areas and at different levels. In the individualized intervention program, the problematic area of the infant is determined according to these evaluations, and the area that has a problem in that area is intervened. These principles are: providing sensory opportunities, posing just right challenges, avoiding negative experiences, cooperating in activity choices, helping self-organization, supporting with the optimum stimuli, creating a play context, maximizing the child’s success, ensuring physical safety, arranging the child’s play environment, and providing an alliance during treatment. All interventions were applied face-to-face individually in therapy rooms in accordance with the sensory integration room plan. Evaluations were administered in both groups before and after the intervention.

Physical therapy intervention was designed according to the infants’ AIMS scores. In the AIMS, on the other hand, there is a development range that should be according to the total evaluation score for each age. Infants whose scored outside this range are considered developmentally intellectually disabled. The AIMS consists of 58 items that evaluate the infant’s movement, and each item is in a sequence towards the level of development. In the AIMS, each item corresponds to a position or a movement pattern. As the infant’s score increases, he can perform more positions, movements and advanced skills. It is not possible to do muscle testing on infants. With the AIMS, we first evaluated the infant, then we determined whether there was a developmental delay. If there was a developmental delay, we created an exercise program for supporting the neurodevelopmental process that allows that movement to take place according to which items the infant can and cannot do. In this way, we determined which muscle groups were insufficient in the trunk, lower and upper extremities, and applied the appropriate exercise intervention. The intervention protocol included prone and supine positioning, rotation, supported and unsupported sitting on a chair, supported and unsupported long sitting, standing, balance exercises on an exercise ball, and functional reaching for toys and objects. The interventions were finalized complying with the TIDieR checklist. TIDieR is a checklist that provides information about the way the intervention was administered in a systematic way. It consists of 11 items and these items provide the details of the intervention plan within certain principles. This ensures that the intervention can be implemented by other practitioners [28].

### 2.4. Data Analysis

Statistical analyses were performed using the Statistical Package for the Social Sciences 25.0 for Windows (SPSS). The normality of the data was analysed with the Kolmogorov–Smirnov test. Descriptive statistics included frequency and percentage for nominal data and mean and standard deviation for quantitative data. Significance level was accepted as *p* < 0.05 at the 95% confidence interval. The categorical variables differences across the groups were analysed with the chi-square test. It was found that the measurement results did not show normal distribution, and thus, non-parametric tests were used. The Mann–Whitney U Test was used to compare the two groups in terms of numerical data. The pre- and post-intervention results of the groups were analysed with the Wilcoxon signed-rank test. Quade’s rank analysis of covariance test was used to compare the post intervention means when assessing baseline differences. We calculated the Cohen’s coefficient d as the effect size of the differences between participants in the intervention and control groups. Effect size benchmarks were determined as 0.20, 0.50, and 0.80 standard deviations and were considered small, medium, and large, respectively [29]. All test and subtest scores were analysed using these analysis methods.

## 3. Results

Thirty-four infants with CP and CVI, including 17 in the intervention group and 17 in the control group, were included in the study. The mean age of the intervention group was 14.47 ± 1.28 months (range: 12–16 months), and it was 13.82 ± 1.55 months in the control group (range: 12–16 months) (*p* > 0.05). The groups were comparable in terms of demographic characteristics (*p* > 0.05) (Table 1).

Pre-intervention TSFI and AIMS scores did not differ significantly between groups (*p* > 0.05) (Table 2).

Comparisons of pre- and post-intervention scores showed significant changes in TSFI and AIMS scores in both groups (*p* < 0.05). Quade’s rank analysis of covariance test was performed to assess the pre-intervention differences. In the intervention group, post-treatment TSFI values, except the oculomotor control value, were significantly different from the baseline values (*p* < 0.05). Both groups were similar in terms of the post-intervention AIMS scores compared to the baseline values (*p* > 0.05) (Table 3).

Post-intervention TSFI and AIMS scores and comparisons between the groups are shown in Table 4. All scores except oculomotor control subdimension of TSFI and AIMS scores differed statistically between the groups (*p* < 0.05). According to the AIMS score, there was no improvement in motor skills with sensory integration therapy (Table 4).

## 4. Discussion

In this single-blind randomized controlled trial, we divided the infants with CVI and CP into two groups and applied weekly two sessions of conventional physiotherapy for 8 weeks in both groups. In addition, we applied sensory integration training in one group for 8 weeks, 2 sessions a week. As a result of these interventions, we observed that sensory processing processes were developed better in infants who received sensory integration training in addition to conventional physiotherapy. In the current study, we clearly demonstrated that the sensory processes of the infants with CVI were affected. Considering that sensory processing was affected, it was also seen that the sensory processes of these infants could develop with sensory integration intervention. The study will guide clinicians and academics who want to conduct scientific studies in this field, in terms of understanding the importance of sensory processing assessment and intervention when evaluating infants with CVI and raising awareness in families about these issues. In addition, the sensory integration intervention in addition to conventional physiotherapy intervention had no effect on motor and oculomotor skills when compared with conventional physiotherapy programs.

### 4.1. Sensory Processing

Infants with CVI were less exposed to sensory stimuli due to vision loss, and both body and environmental awareness may be insufficient; accordingly, sensory problems may be seen in individuals with CVI [14,30]. It can be thought that coordination and sensory integration may be insufficient in children with CVI compared to their peers due to visual loss, and thus, infants are exposed to visual, auditory, tactile, and vestibular stimuli less; however, providing sensory support can compensate for this deficiency [31].

The efficiency of sensory processing interventions in infancy are well documented in the literature [26,32,33]. Previous studies have also shown the effectiveness of sensory integration interventions in children with CP [31,34]. Pekçetin et al., as a result of an 8-week sensory integration intervention, found improvement in sensory areas in the evaluation made with TSFI in premature infants [26]. We found similar results in infants with CVI and CP. InfaSPnts with CP and CVI may have problems in the processing of visual and vestibular input. The impaired processing of visual and vestibular information may cause postural control deficits in children with CP as these sensory information sources are processed in postural control modulation [35]. This deficiency causes insufficiency in coordinated and balanced movement ability against stimuli. In a vestibular sensory intervention study in 4–6 year old children with CP, vestibular skills were found to be increased [31]. In this study, vestibular sensory deficit was developed with vestibular stimulus in infants. In the meta-analysis study examining the intervention studies of sensory integration in infants with CP, 14 studies were examined, and it was found that sensory integration had an effect on sensory, tactile, balance, gross, and fine motor skills and emotional development. In addition, it was found that the effectiveness of the intervention increased as the age of the intervention group decreased. It was stated that the presence of control groups in the studies and the sensory integration studies conducted in a randomized controlled manner revealed more evidence-based results, and it was stated in the literature that studies in younger age groups and with more participation would better explain the effectiveness of sensory integration in children and infants with CP [19].

To the best of our knowledge, this is the first study showing the effect of sensory integration interventions in infants with CP and CVI. All TSFI subheadings except oculomotor control significantly increased with sensory integration intervention. There are studies examining the response to tactile stimuli in children with CP [36,37]. The present results revealed that children with CP had problems in tactile stimulus modulation. It was emphasized that some children could not respond to stimuli, and some children had a very high response to stimuli and had a very low tolerance to tactile stimuli. In studies conducted on tactile stimulus modulations, the optimal response to the stimulus was achieved with intervention programs [38]. When the tactile stimulus responses were examined in our study, we reached the conclusion that the infants had problems in modulation. With our sensory integration intervention, the response to the tactile stimulus improved in our intervention group.

Oculomotor skills are a very common skill loss with CP [39]. In the sensory integration intervention study, which evaluated oculomotor skills in preterm infants and included oculomotor exercises, improvement was achieved in oculomotor skills at the end of 8 weeks [26]. The fact that visual skills were affected in infants with CVI, unlike preterm infants, may not have been sufficient for the 8-week intervention to develop oculomotor skills.

Infants with CP had problems in responding to stimuli normally and adaptively. The addition of CVI symptoms to CP causes a decrement in visual stimulus inputs. Difficulty perceiving sensory stimuli and impaired modulation in these infants lead to abnormal adaptive responses [6,40]. In our study, we detected inadequacy in adaptive motor responses and provided adaptive motor regulation by providing appropriate to sensory stimulus input.

Our findings may indicate that sensory integration interventions contribute to the development of infants and children in different developmental areas, and that the effect may change when applied more homogeneously and at different times in more specific groups and age categories. Therefore, there is a need for further studies involving different variations. The current study involving infants with CVI and CP provide evidence for efficacy of sensory integration interventions.

### 4.2. Motor Development

Conventional physiotherapy approaches are effective for increasing motor development during childhood [15,41,42,43,44]. However, the sensory integration effect on the motor development of infants with CVI and CP has not been investigated. Our findings provided evidence for the efficiency of sensory integration intervention on motor development. However, the findings of the current study should be supported with further prospective studies.

Previous studies have shown that motor developmental delay due to motor and proprioceptive loss was observed in infants with cortical vision loss [10]. According to our study results, motor development delay may be observed in infants with cortical vision loss, but the sensory integration intervention alone cannot contribute to a significant improvement in motor development.

In a study conducted in children with CP, it was found that bilateral coordination, speed and dexterity of the upper extremities, visual and spatial perception, visual–motor organization and tactile sensory impairments negatively affected handwriting skills, as well as proprioception disorder in the non-hemiplegic side. As another important result, it was emphasized that in treatment approaches in children with hemiplegic cerebral palsy, comprehensive sensory–perceptual–motor evaluations, including both extremities, should be performed in detail at the earliest possible stage in order to minimize the existing problems with early treatment policies [45].

In a randomized controlled study conducted in 24 children with CP aged 2–6 years, one group was given sensory integration therapy and the other group a home program. Although motor development was observed in both groups, when the two groups were compared, it was seen that there was a significant difference in gross motor skills, such as sitting and standing, no difference was observed in advanced skills, such as rolling and walking, and sensory integration and vestibular stimulation were found to be important for motor development in children with CP. As a result, in our study, there was no significant difference in motor development in the two groups at the end of the 8 weeks. Further studies examining its effects on motor development in different age groups and intervention times will provide a better understanding of the effectiveness of the treatment [46]. In the study conducted with 30 children with CP, only conventional physiotherapy was given to one group and sensory integration therapy was given to the other group together with conventional physical therapy exercises. Before and after the intervention, motor and sensory evaluations were made in all participants in both groups, and exercises were given to the groups 5 days a week for 6 months. As a result, it shows that when sensory integration therapy and conventional physical therapy exercises are given together, it is more effective and powerful in improving gross motor functions in children with CP rather than just giving traditional physical therapy exercises [47]. When we compare it with our study, both the duration of 8 weeks and the fact that our cerebral palsy group has cortical vision loss explains the difference in terms of the effect of sensory integration intervention on motor development.

In a recent study, 28 patients with CP aged 0–6 years were divided into two groups and an individualized neurodevelopmental treatment approach was applied in one group and sensory integration treatment was applied in the other group for 12 weeks. Proprioception parameters and balance were evaluated with the Pedalo Sensamove Balance Test and motor skill level was assessed with the Gross Motor Function Measure (GMFM-88). After the intervention programs, positive changes were observed in the functionality, sitting balance, motor level, balance, and proprioception measurements of the subjects in both groups, but there was no statistical significance between the groups. Although no significant difference was found when the two therapy programs were compared in children with cerebral palsy, positive changes were noted in in-group comparisons. In this context, the necessity of including structured sensory integration practices in the individualized education programs in children with CP was revealed in the study [48]. A 12-week sensory integration training may be an alternative treatment for motor development; however, further studies are needed to support this claim.

Our study findings indicated that infants with CP and CVI may have problems in sensory areas, such as motor and tactile, vestibular, proprioception and visual. As a result of the intervention, it was observed that the 8-week sensory integration intervention improved sensory processing except oculomotor skills in infants with CVI, but this intervention did not provide an improvement in motor skills. In future studies, investigating the effect of sensory integration on motor development with a longer-term intervention program may provide a clearer understanding of the relationship between sensory and motor development in infants with CVI. The fact that we performed the sensory integration intervention on infants in our study and that we obtained meaningful results enabled us to demonstrate how important early intervention is. We recommend that physiotherapists, occupational therapists, and clinicians working in the field of rehabilitation with low vision definitely include infants in sensory integration evaluation and inform families in order to raise awareness in this regard.

## 5. Limitation

The main limitation of the current study was that the infants participating in the study were not homogenous in terms of the type of CP and gender distribution. Infants with different types of CP may cause differences in results because different CP types exhibit different symptoms. Further studies should be carried out in a homogeneous sample in this regard. The fact that the duration of the intervention was limited to 8 weeks may be considered insufficient to reveal the effect of sensory integration on motor development. Further studies should evaluate sensory integration therapy effect on motor development with longer interventions.

## Figures and Tables

**Figure 1 children-09-01123-f001:**
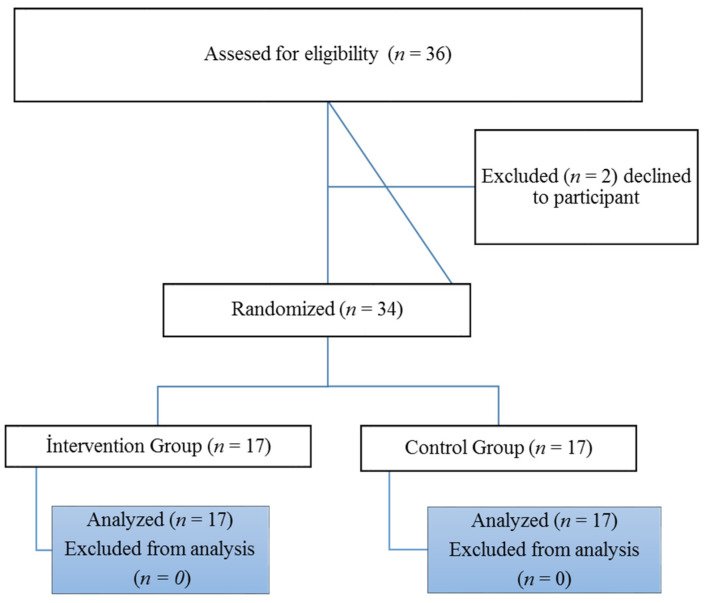
Flow diagram: the flow of during each trial phase.

**Table 1 children-09-01123-t001:** Demographic Characteristics of Groups.

		Intervention Group		Control Group		*p*
		*n* = 17	%	*n* = 17	%	
Gender	Female	5	29.4%	11	64.7%	0.7154
	Male	12	70.6%	6	35.3%	
Gestational week	Less than 37 weeks	9	52.9%	10	58.8%	0.730
	37 weeks and later	8	47.1%	7	41.2%	
Diagnosis	Spastic CP	0	0%	3	17.7%	0.147
	Dystonic CP	4	30.7%	5	29.4%	
	Hypotonic CP	13	69.3%	9	52.9%	

SD: Standard deviation, Chi-Square test.

**Table 2 children-09-01123-t002:** Comparison of Pre-Treatment Scores in the Test of Sensory Functions in Infants and the Albert Infant Motor Scale.

	Intervention Group (*n* = 17)	Control Group (*n* = 17)	*p*
	Mean ± SD	Mean ± SD	
Response to tactile deep pressure	2.94 ± 1.08	2.94 ± 0.96	0.986
Adaptive motor functions	3.52 ± 1.37	3.82 ± 1.23	0.423
Visual–tactile integration	3.47 ± 1.37	2.94 ± 1.08	0.296
Oculomotor control	0.70 ± 0.58	0.41 ± 0.50	0.141
Response to vestibular stimuli	3.41 ± 1.22	3.52 ± 1.,12	0.886
TSFI total score	14.05 ± 4.22	13.64 ± 3.49	0.809
AIMS	19.64 ± 8.14	13.76 ± 7.20	0.054

SD: Standard deviation, AIMS: Albert Infant Motor Scale, TSFI: Test of Sensory Functions in Infants.

**Table 3 children-09-01123-t003:** Comparison of intra-group pre-treatment and post-treatment evaluations.

	Intervention Group				Control Group				Between-Group Comparison		
	Baseline Assessment Mean ± SD	Final Assessment Mean ± SD	*p*	Cohen’s d	Baseline Assessment Mean ± SD	Final Assessment Mean ± SD	*p*	Cohen’s d	F	*p*	d
Response to tactile deep pressure	2.94 ± 1.08	6.35 ± 1.41	<0.001	3.139	2.94 ± 0.96	3.94 ± 0.96	<0.001	1.877	81.48	<0.001 *	0.691
Adaptive motor functions	3.52 ± 1.37	7.82 ± 2.78	<0.001	2.464	3.82 ± 1.23	4.70 ± 1.26	<0.001	2.383	67.52	<0.001 *	0.595
Visual–tactile integration	3.47 ± 1.37	6.64 ± 1.36	<0.001	2.959	2.94 ± 1.08	3.52 ± 1.17	<0.001	0.902	107.21	<0.001 *	0.734
Oculomotor control	0.7 ± 0.58	1.52 ± 0.51	<0.001	2.238	0.41 ± 0.5	1.41 ± 0.71	<0.001	2.351	1.50	0.228	0.039
Response to vestibular stimuli	3.41 ± 1.22	6.94 ± 1.81	<0.001	3.521	3.52 ± 1.12	4.35 ± 1.45	<0.001	0.913	62.77	<0.001 *	0.670
TSFI total score	14.05 ± 4.22	29.29 ± 6.84	<0.001	4.654	13.64 ± 3.49	17.94 ± 3.49	<0.001	2.138	199.53	<0.001 *	0.759
AIMS	19.64 ± 8.14	20.94 ± 8.79	<0.001	1.213	13.76 ± 7.20	14.94 ± 8	<0.001	1.226	1.22	0.276	0.003

SD: Standard deviation, AIMS: Albert Infant Motor Scale, TSFI: Test of Sensory Functions in Infants * *p* < 0.05.

**Table 4 children-09-01123-t004:** Means of TSFI and subheading scores, AIMS score, and eye movement angles and post-treatment comparisons between groups.

Variable	Intervention Group (*n* = 17)	Control Group (*n* = 17)	*p*
	**Mean ± SD**	**Mean ± SD**	
Response to tactile deep pressure	6.35 ± 1.41	3.94 ± 0.96	<0.001 *
Adaptive motor functions	7.82 ± 2.78	4.70 ± 1.26	<0.001 *
Visual–tactile integration	6.64 ± 1.36	3.52 ± 1.17	<0.001 *
Oculomotor control	1.52 ± 0.51	1.41 ± 0.71	0.755
Response to vestibular stimuli	6.94 ± 1.81	4.35 ± 1.45	<0.001 *
TSFI total score	29.29 ± 6.84	17.94 ± 3.49	<0.001 *
AIMS	20.94 ± 8.79	14.94 ± 8	0.078

SD: Standard deviation, Mann–Whitney U Test * *p* < 0.001, AIMS: Albert Infant Motor Scale, TSFI: Test of Sensory Functions in Infants.

## Data Availability

The dataset analysed in this study can be requested from Mustafa Cemali (muscemali@hotmail.com) on reasonable request.

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
