# Peer review of "The Effectiveness of Sensory Integration Interventions on Motor and Sensory Functions in Infants with Cortical Vision Impairment and Cerebral Palsy: A Single Blind Randomized Controlled Trial"

_children, 2022, doi:10.3390/children9081123_

Round 1

Reviewer 1 Report

The introduction and justification for the study is appropriate

The methodology is correct in its overall design, although some aspects are mentioned below:

- The manuscript states that the physiotherapist was blinded to both group distribution and study design. However, it seems from the characteristics of the physiotherapist that it was the same physiotherapist who applied the treatment and assessed the children. How is this possible? Explain further

- It states that the physiotherapy intervention was delivered on the basis of the children's AIMS scores. Further specify the intervention differences in terms of AIMS scores.

- In the intervention group, specify whether the two weekly sessions of conventional physiotherapy were on the same days or on other sensory integration days.

The results are clearly reflected

The discussion adequately analyses and justifies the results found.

Author Response

July 11, 2022

First of all, we would like to thank you and all the referees for their suggestions/comments, which have helped to improve our manuscript. We have attempted to address all of the issues brought up in suggestions by revising the manuscript as needed.

You may find our responses to the referees’ comments below. Revisions are shown in green highlight below our responses when appropriate.

Reviwer 1

Thank you for your valuable comments. We made the necessary arrangements in line with your suggestions.

1. The manuscript states that the physiotherapist was blinded to both group distribution and study design. However, it seems from the characteristics of the physiotherapist that it was the same physiotherapist who applied the treatment and assessed the children. How is this possible? Explain further

R. We made a mistake, I apologize, we took your warning into account and made the necessary arrangements.

All interventions and assesments were performed by a physiotherapist who worked in the field of physiotherapy for 8 years and occupational therapy for 6 years, blind to assessment results. This physiotherapist has a master's degree in both fields and is currently pursuing a doctorate in occupational therapy.

2. It states that the physiotherapy intervention was delivered on the basis of the children's AIMS scores. Further specify the intervention differences in terms of AIMS scores.

R. We made the necessary additions in line with the suggestions:

In the AIMS, each item corresponds to a position or a movement pattern. As the infant's score increases, he can perform more positions, movements and advanced skills. It is not possible to do muscle testing on infants. With the AIMS, we first evaluated the infant, then we determined whether there was a developmental delay. If there was a developmental delay, we created an exercise program for supporting the neurodevelopmental process that allow that movement to take place according to which items the infant can and cannot do. In this way, we determined which muscle groups were insufficient in the trunk, lower and upper extremities and applied the appropriate exercise intervention.

3. In the intervention group, specify whether the two weekly sessions of conventional physiotherapy were on the same days or on other sensory integration days.

R.We made the necessary additions in line with your suggestions. Sensory integration therapy was performed on 2 separate days in the same week that there was no physical therapy session.

Reviewer 2 Report

Summary:

The authors observed that infants with Cerebral Palsy with comorbid CVI may not receive adequate therapy for sensory integration to compensate for sensory engagement lost due to the CVI. Thus the authors compared the addition of sensory integration intervention to conventional therapy with conventional therapy alone to test whether this would increase sensory-motor skills on relevant clinical batteries. The authors show in a single blinded RCT that on nearly all TSFI tasks, the intervention therapy does improve sensory test performance beyond conventional therapy alone. Motor skills, as shown by AIMS test results, are not significantly improved by the intervention therapy over the conventional therapy alone.

The execution of statistical methods is a strength of this paper. The experimental study was well formulated to test the paper’s main research question, and the results were reported clearly. Overall, this was a clever trial that addressed a gap in current therapy and research attentions for CP and CVI. The reviewer recommends the article be accepted after minor revisions.

Review:

There are a few significant areas in need of clarification or revisions throughout the paper. These pertain primarily to:

  • A need for a thorough explanation or rationale for the number of therapy sessions per week for each group. The report states that the intervention group received two sensory integration sessions in addition to the conventional sessions whereas controls received only two conventional sessions. The authors should include the rationale for the use of two vs four conventional sessions for the control group, and an explanation of why the effect seen may be or is not due to two extra therapy sessions per week for the intervention group.

  • It is not clear whether the results are extended to motor skills as well as sensory test improvements. The AIMS score would suggest motor skills are not benefited, and the abstract states only sensory improvement, but other places in the introduction and discussion may be suggesting motor skills are also improved further by the sensory integration therapy.

  • The reviewer noted and agrees with the limitations that the authors state as such.

  • The discussion of prior work in the introduction and discussion sections could be improved. Material is introduced in the sensory processing and motor skills discussions not introduced in the introduction, and its reason for inclusion is unclear. A better thread to tie together the results of reviewed articles used as supporting literature and further emphasis of the point the inclusion the discussion of those papers’ findings would be good. Particular attention should be given to studies that testing very similar parameters (59 – 64, discussion 100 -119).

  • The closing lines of the introductory section should include a more clearly stated hypothesis or research question for the study.

Minor issues:

12: CVI and CP?

25 -26: To “increasing sensory processing skills” add “in one measure or test” to capture the non significant AIMS results.

40 – The range in cortical vision loss percentages reported is large. It warrants a comment on the spread or multiple sources showing how it varies between studies.

144-154 – Lines are difficult to follow and need to be revised for clarity.

Paragraph beginning 144 - Briefly discuss conventional control therapy methods in addition to the intervention therapy explanation.

157 – “individualized sensory integration…” needs clarification and explanation. Were sessions different for each individual treated? How might that impact results?

A non-exhaustive list of lines in need of clarifying or grammatical corrections are needed are below, and it is assumed the will be addressed by final editorial review.

22, 31, 45-47, 62, 64, 65, 73

Author Response

July 11, 2022

First of all, we would like to thank you and all the referees for their suggestions/comments, which have helped to improve our manuscript. We have attempted to address all of the issues brought up in suggestions by revising the manuscript as needed.

You may find our responses to the referees’ comments below. Revisions are shown in yellow highlight below our responses when appropriate.

Reviewer 2:

Thank you for your valuable comments. We had a hard time writing this article due to the limited literature. Since there is no such study in infants with CVI, we could not discuss similar studies. We used many of the studies that we found after a wide literature review, which are close to the subject, in the introduction and discussion. In this sense, we hope that our study will provide a perspective for similar studies.

Review

1. A need for a thorough explanation or rationale for the number of therapy sessions per week for each group. The report states that the intervention group received two sensory integration sessions in addition to the conventional sessions whereas controls received only two conventional sessions. The authors should include the rationale for the use of two vs four conventional sessions for the control group, and an explanation of why the effect seen may be or is not due to two extra therapy sessions per week for the intervention group.

R. We made the necessary additions and explanations in line with your suggestions:

“The intervention program includes conventional physiotherapy and sensory integration training. Both control and intervention groups were applied physiotherapy intervention as 2 sessions of 45 minutes per week for 8 weeks. In addition to the physiotherapy intervention, the intervention group received sensory integration intervention as 2 sessions of 45 minutes per week for 8 weeks. Sensory integration therapy was performed on 2 separate days in the same week that there was no physical therapy session. Conventional physiotherapy applied to both groups included classical physiotherapy such as rotation, sitting without support, standing and balance and strengthening exercises. Sensory integration therapy applied only to the intervention group included interventions involving vision, hearing, touch, and vestibular stimuli.

The intervention duration and protocol were designed according to the one of our previous works [26].

Sensory integration therapy is a treatment that includes sensory stimuli, not a physiotherapy intervention. In the literature, sensory integration is applied by clinicians as a sensory-based supportive treatment in addition to physiotherapy sessions. Therefore, we cannot say that we expect a satisfactory recovery from the intervention group receiving more sessions.

2. It is not clear whether the results are extended to motor skills as well as sensory test improvements. The AIMS score would suggest motor skills are not benefited, and the abstract states only sensory improvement, but other places in the introduction and discussion may be suggesting motor skills are also improved further by the sensory integration therapy.

R. Thank you for highlighting this issue, we would like to inform you about an issue that needs clarification. We've added this information to the results section:

“According to the AÄ°MS score, there was no improvement in motor skills with sensory integration therapy.

There are studies in the literature showing that sensory integration intervention has an effect on motor skills. There may have been differences in the studies, since the intervention groups, age and duration of the application varied. The fact that we carried out our study especially in infants with CVI may also make a difference. We planned this study to see these differences. We had a hard time finding resources to use in the introduction and discussion. The lack of such a study in this group caused us to write a limited introduction and discussion. We tried to establish relationships with different age categories and groups. In future studies, the effects of sensory integration on both motor and sensory development in infants with CVI will be investigated further and the results will be interpreted in much more detail.

3. The discussion of prior work in the introduction and discussion sections could be improved. Material is introduced in the sensory processing and motor skills discussions not introduced in the introduction, and its reason for inclusion is unclear. A better thread to tie together the results of reviewed articles used as supporting literature and further emphasis of the point the inclusion the discussion of those papers’ findings would be good. Particular attention should be given to studies that testing very similar parameters (59 – 64, discussion 100 -119).

R. We made the necessary arrangements and explanations in line with your suggestions.

Children with CP experience sensory processing difficulties in visual auditory vestibular tactile position when compared with healthy children [6]. In a randomized controlled study conducted with 22 children aged 2-6 years with spastic CP, who did not experience CVI,  “It was found that sensory integration therapy applied 3 days a week for 3 months improves motor skills [7].”

In a randomized controlled study with 26 children aged 2-4 years with CVI and spastic CP, “It was found that sensory integration therapy applied in addition to conventional physiotherapy for 3 days a week for 3 months improved gross and fine motor skills [10].”

“Pekçetin et al. As a result of 8-week sensory integration intervention, he found improvement in sensory areas in the evaluation made with TSFI in premature infant [26]. We found similar results in infant with CVI and CP.

4. The closing lines of the introductory section should include a more clearly stated hypothesis or research question for the study.

R. We made the necessary addition to this part:

“The hypothesis of the study is that sensory integration therapy improves sensory and motor skills in infants with cerebral palsy with cortical visual impairment.”

Minor issues:

1. 12: CVI and CP?

R. Cortical visual impairment is the visual impairment seen in cerebral palsy and is within the scope of Cerebral palsy. Infant with CP have both those with CVI and those without. "with CVI and CP" means cerebral palsy with cortical visual impairment.

2. 25 -26: To “increasing sensory processing skills” add “in one measure or test” to capture the non significant AIMS results.

R. We made the necessary addition to this part:

Sensory integration intervention delivered with the conventional physiotherapy program was more effective than the conventional physiotherapy program in increasing sensory processing skills in one measure in infants with CVI and CP. 

3. 40 – The range in cortical vision loss percentages reported is large. It warrants a comment on the spread or multiple sources showing how it varies between studies.

R. We made the necessary corrections in line with your suggestions:

“Cortical vision loss is seen in 60%-70% of children with CP. One of the most common problems in children with CP is CVI [4].”

4. 144-154 – Lines are difficult to follow and need to be revised for clarity.

R. In this section, we have more clearly expressed the intervention information applied in order to regulate the general flow:

“The intervention program includes conventional physiotherapy and sensory integration training. Both control and intervention groups were applied physiotherapy intervention as 2 sessions of 45 minutes per week for 8 weeks. In addition to the physiotherapy intervention, the intervention group received sensory integration intervention as 2 sessions of 45 minutes per week for 8 weeks. Sensory integration therapy was performed on 2 separate days in the same week that there was no physical therapy session. Conventional physiotherapy applied to both groups included classical physiotherapy such as rotation, sitting without support, standing and balance and strengthening exercises. Sensory integration therapy applied only to the intervention group included interventions involving vision, hearing, touch, and vestibular stimuli.”

5. Paragraph beginning 144 - Briefly discuss conventional control therapy methods in addition to the intervention therapy explanation.

R. We made the necessary additions in line with your suggestions:

“Conventional physiotherapy applied to both groups included classical physiotherapy such as rotation, sitting without support, standing and balance and strengthening exercises.”

6. 157 – “individualized sensory integration…” needs clarification and explanation. Were sessions different for each individual treated? How might that impact results?

R. The rehabilitation session individualized according to measurement results. Sensory integration therapy is planned according to the response of infants to sensory stimuli. Intervention should be preferred in line with the needs of each infant. If we do not apply a personalized intervention program in this way and the same sensory integration intervention is applied to everyone, this intervention may cause us to give more than necessary sensory stimulus to some infant or less than necessary to some infant. At the end of the process intervention can both harm the infant and prevent the correct results..

We made the necessary additions in line with your suggestions: “According to the evaluations, infant may have problems in different sensory areas and at different levels. In the individualized intervention program, the problematic area of the infant is determined according to these evaluations, and the area that has a problem in that area is intervened.”

Reviewer 3 Report

Known in the field based on previous literatures:

  1. Cerebral palsy (CP) is a group of neurological disorder caused by brain damage. It is the most common motor and movement disability of childhood and some of the potential problems a child with CP may face include movement and walking disabilities, spinal and joint problems, speech difficulties, learning disabilities, hearing or vision loss.
  2. Cerebral or cortical visual impairment (CVI), a disorder caused by damage to the parts of the brain that process vision, is diagnosed when children show abnormal visual responses that aren’t caused by the eyes themselves. CVI can cause a variety of visual problems that include seeing certain parts of what is in front of them, like busy moving scenes, recognizing faces and objects.

In this article authors testified following observation:

1. In this article, authors studied patients which were showing the effect of sensory integration interventions in infants with CP and CVI.

2. Authors showed significant improvement in many sensory and motor parameters in intervention group.

Authors nicely mentioned many facts related to CP, CVI and effect of sensory integration interventions.  The mentioned suggestions if incorporated could help in the better understanding of the significance of the study and implications.

Minor Concerns:

1. Authors reported significant improvement in various parameters of intervention group at 8 weeks. In table 4 variables, have you compared the level of significance at 0, 2, and 4 weeks to see the time dependent improvement?

2. Is there any plan to follow up the same patients for longer time to see more improvement?

3. How do you differentiate the level of severeness of CP and CVI in patients and the effect of sensory integration intervention improvement? Is there any effect of gender/ hormones?

4. Please explain the reason as effectiveness of the intervention increased as the age of the intervention group decreased.

5. Is there any other method of improvement apart from sensory integration intervention? if yes, explain little bit about them. 

Author Response

July 11, 2022

First of all, we would like to thank you and all the referees for their suggestions/comments, which have helped to improve our manuscript. We have attempted to address all of the issues brought up in suggestions by revising the manuscript as needed.

You may find our responses to the referees’ comments below. Revisions are shown in green highlight below our responses when appropriate.

Reviewer 3

Thank you for your valuable comments. Each of your comments has allowed us to look at the subject from a very different perspective. We have tried to clarify your questions as much as we can. We would like to plan studies taking into account what you said in future studies.

Minor Concerns:

1. Authors reported significant improvement in various parameters of intervention group at 8 weeks. In table 4 variables, have you compared the level of significance at 0, 2, and 4 weeks to see the time dependent improvement?

R. In our study plan, we only made assessments at the end of 8 weeks before starting 8 weeks of treatment. Since we did not make interim evaluations, we did not compare the significance levels at 2nd and 4th weeks.

2. Is there any plan to follow up the same patients for longer time to see more improvement?

R. We did not make such a plan for this study, but we are considering planning a study to see long-term effects in future studies.

3. How do you differentiate the level of severeness of CP and CVI in patients and the effect of sensory integration intervention improvement? Is there any effect of gender/ hormones?

R. We made your evaluations on motor skills and sensory integration parameters. We performed motor skills with AIMS and sensory, visual and oculomotor skills with TSFI. Apart from this, we did not perform special CP and CVI evaluation. As we compared intervention efficacy in our study, we revealed how much improvement there was in these parameters.

R. We did not use a gender-focused intervention and evaluation method in our study, it would not be correct to comment on this study since both genders were participants. In the literature, there is no clear distinction regarding gender. In future studies, the study can be planned by considering this distinction.

R. I have never seen a study that included the evaluation of these parameters in relation to hormonal level in infants with CP with CVI. There are very few studies, even with a focus on rehabilitation, in infants with CVI and CP. In future studies, the relationship with hormonal level can be examined.

4. Please explain the reason as effectiveness of the intervention increased as the age of the intervention group decreased.

R. Based on the knowledge that plasticity and neuroplasticity are high between 0-3 years of age with the birth of babies, we can say that they are much higher in the first year of birth.

5. Is there any other method of improvement apart from sensory integration intervention? if yes, explain little bit about them.

5. We only apply sensory integration therapy for sensory development. For motor skill development, we apply physiotherapy treatment and sensory integration therapy, which includes sensory stimuli. Due to our area of expertise, there is no other treatment that we can provide this improvement.
